# Combined associations of body mass index and adherence to a Mediterranean-like diet with all-cause and cardiovascular mortality: A cohort study

Karl Michaëlsson[1]*, John A. Baron[1,2,3,4], Liisa Byberg[1], Jonas Höijer[1], Susanna C. Larsson[1,5], Bodil Svennblad[1], Håkan Melhus[6], Alicja Wolk[1,5], Eva Warensjö Lemming[1]

1 Department of Surgical Sciences, Uppsala University, Uppsala, Sweden, 2 Department of Epidemiology, Geisel School of Medicine at Dartmouth, Hanover, New Hampshire, United States of America, 3 Department of Medicine, University of North Carolina School of Medicine, Chapel Hill, North Carolina, United States of America, 4 Department of Epidemiology, Gillings School of Global Public Health, University of North Carolina, Chapel Hill, North Carolina, United States of America, 5 Unit of Cardiovascular and Nutritional Epidemiology, Institute of Environmental Medicine, Karolinska Institutet, Stockholm, Sweden, 6 Department of Medical Sciences, Uppsala University, Uppsala, Sweden

* karl.michaelsson@surgsci.uu.se

**Data Availability Statement:** Data cannot be shared publicly because of the sensitive nature of the data and the GDPR legislation. Data are

## Abstract

### Background

It is unclear whether the effect on mortality of a higher body mass index (BMI) can be compensated for by adherence to a healthy diet and whether the effect on mortality by a low adherence to a healthy diet can be compensated for by a normal weight. We aimed to evaluate the associations of BMI combined with adherence to a Mediterranean-like diet on all-cause and cardiovascular disease (CVD) mortality.

### Methods and findings

Our longitudinal cohort design included the Swedish Mammography Cohort (SMC) and the Cohort of Swedish Men (COSM) (1997–2017), with a total of 79,003 women (44%) and men (56%) and a mean baseline age of 61 years. BMI was categorized into normal weight (20–24.9 kg/m$^2$), overweight (25–29.9 kg/m$^2$), and obesity (30+ kg/m$^2$). Adherence to a Mediterranean-like diet was assessed by means of the modified Mediterranean-like diet (mMED) score, ranging from 0 to 8; mMED was classified into 3 categories (0 to <4, 4 to <6, and 6–8 score points), forming a total of 9 BMI × mMED combinations. We identified mortality by use of national Swedish registers. Cox proportional hazard models with time-updated information on exposure and covariates were used to calculate the adjusted hazard ratios (HRs) of mortality with their 95% confidence intervals (CIs). Our HRs were adjusted for age, baseline educational level, marital status, leisure time physical exercise, walking/cycling, height, energy intake, smoking habits, baseline Charlson's weighted comorbidity index, and baseline diabetes mellitus. During up to 21 years of follow-up, 30,389 (38%) participants died, corresponding to 22 deaths per 1,000 person-years. We found the lowest HR of all-cause

available from the national research infrastructure SIMPLER for researchers who meet the criteria for access to confidential data. Details of how to obtain data from the national research infrastructure SIMPLER can be obtained at the website www. simpler4health.se. Our study has the SIMPLER project reference SIMP2019004.

**Funding:** The study was supported by grants from the Swedish Research Council (https://www.vr.se; grant nos. 2015-03257, 2017-00644, and 2017-06100 to KM). The national research infrastructure SIMPLER receives funding through the Swedish Research Council under the grant no. 2017-00644 (to Uppsala University and KM). The computations were performed on resources provided by the Swedish National Infrastructure for Computing's (www.snic.se) support for sensitive data SNIC-SENS through the Uppsala Multidisciplinary Center for Advanced Computational Science (UPPMAX) under Project SIMP2019004. SNIC is financially supported by the Swedish Research Council. The funders had no role in study design, data collection and analysis, decision to publish, or preparation of the manuscript.

**Competing interests:** The authors have declared that no competing interests exist.

**Abbreviations:** BMI, body mass index; CI, confidence interval; COSM, Cohort of Swedish Men; CVD, cardiovascular; FFQ, food frequency questionnaire; HR, hazard ratio; mMED, modified Mediterranean-like diet; RD, risk difference; RR, relative risk; SMC, Swedish Mammography Cohort; STROBE, Strengthening the Reporting of Observational Studies in Epidemiology.

mortality among overweight individuals with high mMED (HR 0.94; 95% CI 0.90, 0.98) compared with those with normal weight and high mMED. Using the same reference, obese individuals with high mMED did not experience significantly higher all-cause mortality (HR 1.03; 95% CI 0.96–1.11). In contrast, compared with those with normal weight and high mMED, individuals with a low mMED had a high mortality despite a normal BMI (HR 1.60; 95% CI 1.48–1.74). We found similar estimates among women and men. For CVD mortality (12,064 deaths) the findings were broadly similar, though obese individuals with high mMED retained a modestly increased risk of CVD death (HR 1.29; 95% CI 1.16–1.44) compared with those with normal weight and high mMED. A main limitation of the present study is the observational design with self-reported lifestyle information with risk of residual or unmeasured confounding (e.g., genetic liability), and no causal inferences can be made based on this study alone.

## Conclusions

These findings suggest that diet quality modifies the association between BMI and all-cause mortality in women and men. A healthy diet may, however, not completely counter higher CVD mortality related to obesity.

## Author summary

### Why was this study done?

- It is unclear whether the effect on mortality of a higher BMI can be compensated for by adherence to a healthy diet.

- It is also unclear whether the effect on mortality by a low adherence to a healthy diet can be compensated for by a normal weight.

### What did the researchers do and find?

- We conducted a population-based cohort study that included women and men with time-updated lifestyle information.

- Obese individuals with high adherence to a Mediterranean-type diet did not experience the increased overall mortality otherwise associated with high BMI, although higher CVD mortality remained.

- Lower BMI did not counter the elevated mortality associated with a low adherence to a Mediterranean diet.

### What do these findings mean?

- Our results indicate that adherence to healthy diets such as a Mediterranean-like diet may modify the association between BMI and mortality.

High body mass index (BMI) accounted for 4.0 million deaths globally in 2015 [1], and more than two-thirds of these deaths were due to cardiovascular disease (CVD) [1]. At middle age, the lowest mortality rates are found in individuals within the higher range (23.5–24.9 kg/m$^2$) of a normal BMI [2–5], but with increasing age, the nadir in mortality is shifted upwards towards those who are modestly overweight [3]. Despite the increasing prevalence of obesity, the rates of CVD-related death continue to decrease in Western societies [6–9], a trend not explained by medical treatment alone [10, 11]. These observations suggest that other factors might modify the higher risk of CVD associated with higher body mass [12]. Potentially, one such factor is diet [13].

Healthy dietary patterns have been associated with lower disease and mortality rates. Several cohort studies [14–18], a secondary prevention trial [19], and one [20] extensively scrutinized [21] primary prevention trial in high-risk individuals have shown inverse associations between adherence to the Mediterranean and Mediterranean-like diets and CVD risk [18, 22]. Moreover, observational studies [23] and randomized trials of a Mediterranean diet [24–30] have generally found beneficial effects on CVD risk factors that are negatively affected by obesity.

Adherence to healthy diets and a normal body weight are emphasized in current dietary recommendations to prolong life [31]. However, whether the effect on mortality of a higher BMI can be compensated for by adherence to a healthy diet is not known. Likewise, whether the impact on mortality of a low adherence to a healthy diet can be compensated for by a normal weight is unclear. We sought to describe the pattern of mortality with cross-classified categories of healthy eating and BMI. With the recommended high adherence to a healthy diet and normal BMI as the reference, we therefore designed a longitudinal analysis to investigate the combined impact of adherence to a Mediterranean-like diet and BMI on all-cause mortality, with CVD mortality as a secondary outcome.

## Methods

The study population consisted of participants from 2 population-based cohort studies in Sweden: the Swedish Mammography Cohort (SMC) and the Cohort of Swedish Men (COSM), belonging to the national research infrastructure SIMPLER (www.simpler4health.se). The SMC was established in 1987–1990 when women (born 1914–1948, n = 90,303) residing in 2 counties (Uppsala and Västmanland) were invited to a questionnaire survey covering diet and lifestyle, which was completed by 74% of the women. In the fall of 1997, a second extended questionnaire was sent to all SMC participants who were still alive and residing in the study area (n = 56,030). COSM was established in late 1997 when all male residents (n = 100,303) of 2 counties (Örebro and Västmanland) and born between 1918 and 1952 were invited to participate. When compared with the Official Statistics of Sweden, the cohorts well represented the Swedish population in 1997 in terms of age distribution, educational level, prevalence of overweight and obesity, and smoking status [32]. The 1997 questionnaires in both the SMC and COSM were similar except for the sex-specific questions and included almost 350 items that covered life style factors such as body weight and height, diet (using a validated food frequency questionnaire [FFQ]), dietary supplement use, alcohol consumption, smoking, physical activity, sociodemographic data, and self-perceived health status. This questionnaire was completed by 70% of the women and by 49% of the men. Participants with a prior cancer diagnosis or with energy intakes deemed implausible (±3 SDs from the mean of ln-transformed energy intake) were excluded. The final cohorts consisted of 38,984 women in the SMC and 45,906 men in the COSM followed from January 1, 1998. In 2008, a questionnaire covering general

health, lifestyle, and diseases was sent to all participants that had completed the 1997 questionnaire and who were still alive and living in the study area. The response rate was 63% in the SMC and 78% in the COSM. Those who responded to the 2008 questionnaire received an expanded semiquantitative FFQ in 2009; the response rate was 84% and 90% in the SMC and COSM, respectively. This study is reported as per the Strengthening the Reporting of Observational Studies in Epidemiology (STROBE) guidelines (S1 Checklist). The study has ethical approval by the Regional Ethical Review Boards in Uppsala and Stockholm, Sweden. The questionnaires included a written informed consent. A prespecified analysis plan in Swedish can be found at dx.doi.org/10.17504/protocols.io.bgftjtnn and in S1 Protocol in Swedish and with English translation.

## BMI

We categorized BMI into normal (20 to <25 kg/m$^2$), overweight (25 to <30 kg/m$^2$), and obese ($\geq$30 kg/m$^2$), using self-reported weight and height in 1997 and 2008. Overall, 4% of data points were missing. Those with a BMI below 20 kg/m$^2$ at baseline were excluded (n = 3,226, 4%; 2,354 women and 872 men) since a low body mass can reflect frailty or prevalent disease, which were not intended to be examined in this analysis. Therefore, our final data set used for the analyses contains 79,003 women (44%) and men (56%).

## Modified Mediterranean-like diet (mMED) score

The dietary assessment has been described previously [33]. Briefly, the FFQs included 96 and 132 food items in 1997 and 2009, respectively. We calculated an mMED score adapted from the Mediterranean diet scale devised by Trichopoulou and colleagues [34] using previously defined food items [35], but the scoring was modified according to Knudsen and colleagues, rendering a continuous score [36]. Details of the scoring, ranging from 0 to 8 on a continuous scale, are found below; participants with higher score points were more adherent to the diet. For analysis, mMED was classified into 3 predefined categories (0 to <4, 4 to <6, and 6–8 score points) chosen to balance exposure range and numbers of individuals in each category [37].

Participants indicated in the FFQs how often, on average, they had consumed each food item during the last year, choosing from 8 predefined frequency categories ranging from "never/seldom" to "3 or more times per day". Frequently consumed foods such as dairy products and bread were reported as the number of servings per day (open question). Information on fat type including vegetable oils used in cooking and as salad dressing was also reported. At baseline in 1997, 19% of the women and 12% of the men reported use of olive oil in dressing. The corresponding frequencies were 25% and 19% for use of olive oil in cocking. In 2009, 41% of the women and 37% of the men reported use of olive oil in dressing and with similar proportions of olive oil use in cocking. Total amount of alcohol consumed per day was derived from the FFQ by multiplying the reported frequencies with the reported amounts on a single occasion. Energy intake was estimated by multiplying the portion-specific consumption frequency of each food item with the nutrient content obtained from the Swedish food database [33].

The mMED score comprises 8 components: fruit and vegetables (apple, banana, berry, orange/citrus, and other fruit; carrot, beetroot, broccoli, cabbage, cauliflower, lettuce, onion, garlic, pepper, spinach, tomato, and other vegetables), legumes (peas, lentils, beans, and pea soup) and nuts, unrefined or high-fiber grains (whole-meal bread, crisp bread, oatmeal, and bran of wheat), fermented dairy products (sour milk, yoghurt, and cheese), fish (excluding shellfish), red and processed meat, any use of olive or rapeseed oil for cooking or as dressing, and alcohol intake.

An individual with a reported intake above or below a specific cut point for each component of a diet score usually receives discrete score points (0 or 1), but in the method of Knudsen and colleagues [36], each individual receives 1 or a ratio between the actual intake and a chosen intake amount. Such an approach generates continuous component variables and improves precision of the exposure assessment. In the present study, the reference points for fruit and vegetables, legumes and nuts, nonrefined or high-fiber grains, fermented dairy products, and fish were the median intakes in the 1997 data and are lower-intake thresholds. A participant with an intake of legume and nuts of x grams will thus receive the score = $x/\text{median}_{1997}$. For red and processed meat, consumption below the population median intake rendered a score of 1 point, intakes of 2 or more times the population median rendered 0 points, and intakes above the median (but below $2 \times$ population median) rendered a score of $1 - (\text{actual intake} - \text{median}_{1997})/\text{median}_{1997}$. Any use of olive or rapeseed oil gave 1 point and otherwise 0 points. The alcohol component was coded as intake divided by 5 in the range 0–5 grams/day, as 1 in the intake range 5–15 grams/day, as $1 \times (\text{intake} - 15)/15$ in the range above 15 up to 30 gram/day, and 0 for intakes above 30 grams/day.

The same 1997 cutoff points were applied using the 2009 data in order to avoid secular trends and intake differences caused by the fact that the number of food items was higher in the 2009 FFQ. The more detailed FFQ in 2009 is a reflection of a greater diversification of diet over time.

## Assessment of covariates

Covariates obtained from the questionnaires (1997 and 2008/2009) were age, smoking status (including cigarettes per day at different ages), walking/cycling, leisure time, physical exercise during the past year, and, as markers of socioeconomic status, cohabiting/marital status as well as educational level. The exercise questions have been validated against activity records and accelerometer data [38]. Comorbidity, expressed as Charlson's weighted comorbidity index [39, 40], was defined using ICD diagnosis codes (versions 8, 9, and 10) from the National Patient Register from 1964 to before baseline 1 January 1998. Information on diabetes mellitus was retrieved from the questionnaire and from the National Patient Register.

## Assessment of deaths

All-cause mortality was our primary outcome, with information obtained from the continuously updated Swedish Total Population Register. A complete linkage with the register is possible since all Swedish residents have a unique personal identity number. Since 1952, the National Board of Health and Welfare has maintained information with yearly updates on the causes of death for all Swedish residents in the Cause of Death Registry. We used the underlying cause of death to define our secondary outcome, mortality from CVD (ICD-10 codes I00–I99).

## Statistical analysis

For each participant, follow-up time accrued from 1 January 1998 until the date of death, a questionnaire response indicating a BMI $<20$ kg/m$^2$ in 2008 (n = 1,824), or the end of the study period (31 October 2018 for all-cause mortality and 31 December 2017 for CVD mortality). The associations of mMED and BMI with all-cause mortality and CVD mortality were assessed as age and multivariable-adjusted hazard ratios (HRs) with 95% confidence intervals (CIs) by Cox proportional hazards regression models, with time-updated information of all variables except Charlson's comorbidity index and diabetes mellitus (defined only at baseline) and calendar date as the timescale. Both exposures were initially treated as continuous

variables. To select suitable covariates for the multivariable model, we used current knowledge and a directed acyclic graph [41], presented as S2 Protocol. The overall model included sex, age (splines with 3 knots), educational level ($\leq$9, 10–12, >12 years, other), living alone (yes or no), leisure time exercise during the past year (<1 h/w, 1 h/w, 2–3 h/w, 4–5 h/w, >5 h/w), walking/cycling (almost never, <20 min/d, 20–40 min/d, 40–60 min/d, 1–1.5 h/d, >1.5 h/d), height (splines with 3 knots), energy intake (splines with 3 knots), smoking habits (current, former, never), Charlson's weighted comorbidity index (continuous), and diabetes mellitus as a separate marker variable (yes/no). Missing data were imputed (20 imputations) using Stata's "mi" package (multiple imputations using chained equations). The proportion of missing data in the cohorts was 4% for BMI, 3% for height, 9% for walking/bicycling, 11% for exercise, and 6% for marital status. For all other covariates, the percentage of missing was less than 2%. Missingness of foods at baseline was for fruit and vegetables 0.1%, legumes and nuts 3.4%, grains 0.9%, fermented dairy products 2.9%, fish 1.1%, meat 0.7%, olive or rapeseed oil 0%, and alcohol intake 0%.

Nonlinear trends of mortality were assessed using restricted cubic splines with 3 knots placed at centiles 10, 50, and 90 of mMED and BMI, respectively. We performed stratified analyses in subgroups of potential confounders in which BMI below or above the median of 26 kg/m$^2$ and mMED score were examined as continuous variables. The purpose of these analyses in homogeneous strata was 2-fold: to visualize and evaluate potential confounding, although with a limitation of different baseline hazards in the strata, and to evaluate potential effect modification of the exposures.

Combinations of the categories of BMI and mMED were used to jointly classify study participants into 9 strata. Participants with normal BMI and in the highest category of mMED were used as the reference category in these analyses. Test for homogeneity of HRs across strata was done according to Fleiss [42]. We conducted a stratified analysis by sex with all-cause mortality as outcome. A complementary analysis of risk differences (RDs) was suggested by one of the reviewers. By using the approach described by Austin [43], multivariable-adjusted RDs and relative risks (RRs) were calculated from the predicted survival curve based on the Cox model for all-cause mortality. For the main analysis, the RDs and RRs (with 95% CIs based on 500 bootstrap replicates) were calculated at 20 years, and for the sensitivity analysis starting follow-up in 2009, RDs and RRs at 9 years of follow-up were calculated. For the sensitivity analysis, in which no variable information was updated, yet another method based on pseudo-observations [44], as suggested by the reviewer, was used to estimate the RDs.

Additional sensitivity analyses were conducted, excluding those with a BMI greater than 35 kg/m$^2$, restricting those with normal BMI to 22–25 kg/m$^2$, adding adjustment for pack-years of smoking, restricting analysis to never-smokers, excluding those with pre-existing diseases before baseline (chronic obstructive lung disease, cancer, myocardial infarction or other ischemic heart disease, heart failure, peripheral arterial disease, and stroke as suggested by the reviewer), and excluding the first 2 years of follow-up.

Statistical analyses were carried out in Stata version 15.1 (StataCorp, College Station, TX, USA) and in R, version 4.0 (R Core Team, 2020).

## Results

Age-standardized baseline characteristics (mean age 61 years, range 45–83) within categories of BMI and mMED are displayed in Table 1. Ten percent of the participants were obese, and 46% had a normal BMI; 44% of the participants reported dietary habits consistent with high adherence to mMed and 8% low adherence. Individuals who were overweight or obese reported lower educational attainment, a higher prevalence of diabetes, and less exercise than

**Table 1. Age-standardized baseline characteristics of the participants by 3 categories of BMI and 3 categories of Mediterranean diet score, respectively.**

| | | BMI (kg/m$^2$) | | | Mediterranean Diet Score (Score Points) | | |
|---|---|---|---|---|---|---|---|
| | | 20.0–24.9 | 25.0–25.9 | 30 or more | 0 to <4 | 4 to <6 | 6–8 |
| | | n = 36,065 | n = 31,850 | n = 8,129 | n = 6,256 | n = 38,063 | n = 34,684 |
| Female, n (%) | | 17,930 (49.7%) | 12,124 (38.1%) | 3,808 (46.8%) | 2,429 (38.8%) | 15,527 (40.8%) | 16,326 (47.1%) |
| Male, n (%) | | 18,135 (50.3%) | 19,726 (61.9%) | 4,321 (53.2%) | 3,827 (61.2%) | 22,536 (59.2%) | 18,358 (52.9%) |
| Age, mean (SD) | | 61 (9) | 61 (9) | 61 (9) | 61 (10) | 61 (9) | 61 (9) |
| Education, n (%) | <10 years | 24,272 (67.6%) | 23,534 (74.3%) | 6,455 (79.8%) | 5,175 (83.6%) | 29,002 (76.7%) | 22,502 (72.1%) |
| | 10–12 years | 4,227 (11.8%) | 3,458 (10.9%) | 691 (8.6%) | 478 (7.7%) | 3,902 (10.2%) | 4,273 (12.4%) |
| | >12 years | 7,415 (20.6%) | 4,683 (14.8%) | 940 (11.6%) | 537 (8.7%) | 4,929 (13.0%) | 7,785 (22.5%) |
| Height, cm (SD) | | 171 (9) | 172 (9) | 170 (9) | 171 (9) | 172 (9) | 172 (9) |
| BMI (kg/m$^2$) | | 23 (1.3) | 27 (1.4) | 33 (2.8) | 26 (3.8) | 26 (3.5) | 25 (3.3) |
| BMI categories (kg/m$^2$) | 20–24.9 | na | na | na | 2,392 (41.6%) | 16,419 (45.0%) | 17,234 (51.0%) |
| | 25.0–29.9 | na | na | na | 2,434 (42.3%) | 15,797 (43.3%) | 13,654 (40.4%) |
| | 30 or more | na | na | na | 929 (16.1%) | 4,298 (11.8%) | 2,926 (8.7%) |
| Living alone, n (%) | | 6,714 (20.0%) | 5,548 (18.4%) | 1,793 (23.6%) | 1,822 (31.1%) | 7,516 (20.9%) | 5,355 (16.5%) |
| Energy intake, kcal/day (median, IQR) | | 2,086 (1,620, 2,702) | 2,156 (1,659, 2,775) | 2,041 (1,562, 2,652) | 1,700 (1,221, 2,331) | 2,077 (1,586, 2,713) | 2,222 (1,764, 2,808) |
| Charlson comorbidity index, n (%) | 0 | 31,914 (88.5%) | 27,751 (87.1%) | 6,825 (84.0%) | 5,206 (83.2%) | 33,073 (86.9%) | 30,725 (88.6%) |
| | 1 | 3,822 (10.6%) | 3,745 (11.8%) | 1,181 (14.5%) | 935 (15.0%) | 4,554 (12.0%) | 3,667 (10.6%) |
| | 2 or more | 329 (0.9%) | 354 (1.1%) | 123 (1.5%) | 115 (1.8%) | 436 (1.2%) | 292 (0.9%) |
| Diabetes mellitus, n (%) | No | 34,277 (95.0%) | 29,354 (92.2%) | 7,027 (86.4%) | 5,669 (90.6%) | 35,045 (92.1%) | 32,531 (93.8%) |
| | Yes | 1,788 (5.0%) | 2,496 (7.8%) | 1,102 (13.6%) | 587 (9.4%) | 3,018 (7.9%) | 2,153 (6.2%) |
| Smoking status, n (%) | Current use | 8,957 (25.2%) | 6,974 (22.2%) | 1,755 (21.8%) | 2,042 (33.6%) | 9,481 (25.3%) | 7,046 (20.6%) |
| | Former use | 10,233 (28.8%) | 10,955 (34.9%) | 2,907 (36.2%) | 1,576 (25.9%) | 11,474 (30.6%) | 12,009 (35.1%) |
| | Never use | 16,311 (46.0%) | 13,455 (42.9%) | 3,377 (42.0%) | 2,462 (40.5%) | 16,503 (44.1%) | 15,193 (44.4%) |
| Exercise, n (%) | <1 h/w | 5,875 (18.1%) | 6,061 (21.2%) | 2,188 (30.8%) | 1,731 (34.2%) | 8,087 (23.9%) | 5,000 (15.7%) |
| | 1 h/w | 6,397 (19.8%) | 6,354 (22.2%) | 1,666 (23.5%) | 976 (19.3%) | 7,204 (21.3%) | 6,677 (21.0%) |
| | 2–3 h/w | 11,060 (34.0%) | 9,158 (32.1%) | 1,923 (27.1%) | 1,316 (26.0%) | 10,346 (30.6%) | 11,221 (35.2%) |
| | 4–5 h/k | 4,405 (13.5%) | 3,406 (11.9%) | 656 (9.3%) | 475 (9.4%) | 3,888 (11.5%) | 4,392 (13.8%) |
| | 6 h or more/w | 4,792 (14.7%) | 3,576 (12.5%) | 662 (9.3%) | 570 (11.2%) | 4,290 (12.7%) | 4,563 (14.3%) |
| Walking or cycling, n (%) | Almost never | 3,209 (9.7%) | 3,870 (13.3%) | 1,501 (20.5%) | 1,119 (21.1%) | 4,939 (14.3%) | 2,934 (9.1%) |
| | <20 min/d | 6,461 (19.5%) | 6,866 (23.5%) | 1,889 (25.8%) | 1,183 (22.3%) | 7,971 (23.1%) | 6,680 (20.6%) |
| | 20–40 min/d | 10,992 (33.2%) | 9,083 (31.1%) | 2,001 (27.4%) | 1,436 (27.1%) | 10,360 (30.0%) | 10,978 (33.9%) |
| | 40–60 min/d | 6,097 (18.4%) | 4,672 (16.0%) | 925 (12.6%) | 704 (13.3%) | 5,538 (15.5%) | 5,974 (18.5%) |
| | 1–1.5 h/d | 3,531 (10.7%) | 2,519 (8.6%) | 558 (7.6%) | 408 (7.7%) | 3,212 (9.3%) | 3,247 (10.0%) |
| | >1.5 h/d | 2,848 (8.6%) | 2,154 (7.4%) | 437 (6.0%) | 449 (8.5%) | 2,727 (7.9%) | 2,529 (7.8%) |
| Mediterranean diet score, (median, IQR) | | 5.9 (5.1, 6.7) | 5.8 (5.0, 6.6) | 5.6 (4.7, 6.4) | 3.5 (3.1, 3.8) | 5.3 (4.8, 5.6) | 6.7 (6.3, 7.1) |
| Mediterranean diet score, (units) | 0–4 | 2,422 (6.7%) | 2,427 (7.6%) | 908 (11.2%) | na | na | na |
| | 5 to <6 | 16,439 (45.6%) | 15,794 (49.6%) | 4,288 (52.7%) | na | na | na |
| | 6–8 | 17,204 (47.7%) | 13,629 (42.8%) | 2,933 (36.1%) | na | na | na |

**Abbreviations:** BMI, body mass index; na, not applicable.

those with a normal BMI. Those with high mMED had higher educational attainment, higher physical activity level and energy intake, and a higher prevalence of cohabitation.

During up to 21 years of follow-up (mean 17.4 years) that accrued 1,372,266 person-years of observation, 30,389 (38%) participants died (22 deaths per 1,000 person-years). HRs of

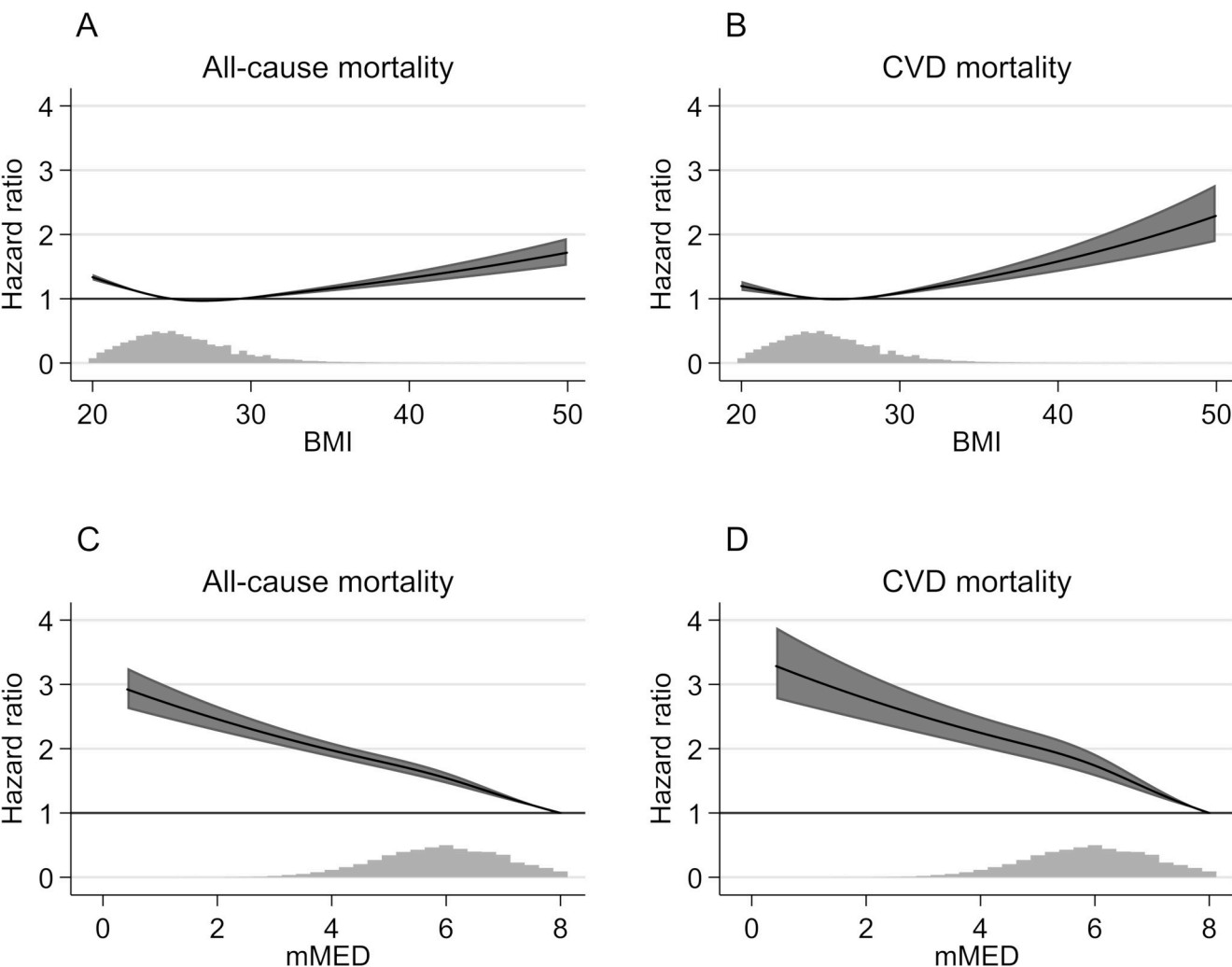

**Fig 1. Association between BMI (A for all-cause mortality and B for cardiovascular mortality) and an mMED score (C for all-cause mortality and D for cardiovascular mortality) with mortality.** The dark gray shaded regions in the figures correspond to 95% CIs, and the spike plots represent the distribution of BMI and mMED scores, respectively. Assessed by multivariable-adjusted HRs using of Cox regression analysis and restricted cubic splines, with a BMI of 25 kg/m$^2$ and mMED score of 8 units as references. HRs adjusted for sex, age (splines with 2 knots), educational level (≤9, 10–12, >12 years, other), living alone (yes or no), leisure time physical exercise during the past year (<1 h/w, 1 h/w, 2–3 h/w, 4–5 h/w, >5 h/w), walking/cycling (almost never, <20 min/d, 20–40 min/d, 40–60 min/d, 1–1.5 h/d, >1.5 h/d), height (splines with 2 knots), energy intake (splines with 2 knots), smoking habits (current, former, never), Charlson's weighted comorbidity index (continuous; 1–16), and diabetes mellitus (yes/no). BMI, body mass index; CI, confidence interval; CVD, cardiovascular disease; HR, hazard ratio; mMED, modified Mediterranean-like diet.

death were related to BMI in a J-shaped pattern (Fig 1A for all-cause mortality and Fig 1B for cardiovascular mortality) and inversely with adherence to mMED (Fig 1C for all-cause mortality and Fig 1D for cardiovascular mortality). The nadir in HRs of all-cause mortality was around a BMI of 26 kg/m$^2$, the median, with an HR of 1.022 (95% CI 1.017–1.027) per 1 kg/m$^2$ above this level. Each unit higher mMED score was associated with a multivariable-adjusted HR of 0.860 (95% CI 0.849–0.871).

Fig 2A and Fig 2B illustrate the analysis of mortality by categories of covariates for BMI as a continuous variable (in subgroups below or above the median of 26 kg/m$^2$ to take into account the J-shaped association with risk) and for mMED score as a continuous variable. The overall pattern of the HRs for all-cause mortality were in the same direction within each subgroup.

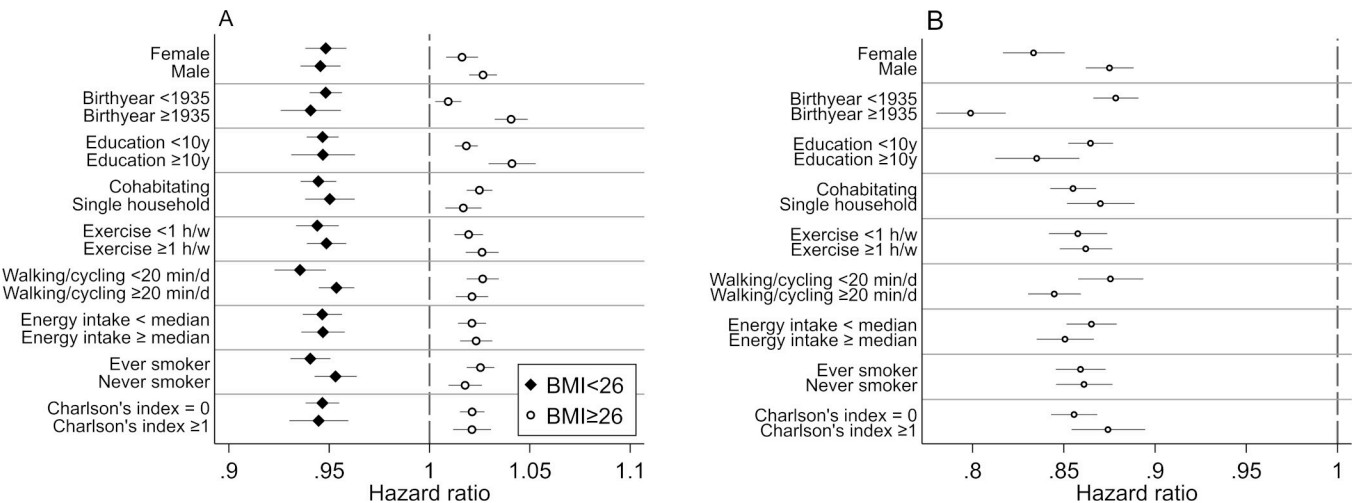

**Fig 2. Subgroup analysis for BMI as continuous variable below or above the median of 26 kg/m² (A) and for mMED (B) as a continuous variable by categories of the covariates.** The whiskers represent 95% CIs. Associations expressed as multivariable-adjusted HRs of all-cause mortality by 1 unit change in BMI or mMED score. HRs adjusted for sex, age (splines with 2 knots), educational level (≤9, 10–12, >12 years, other), living alone (yes or no), leisure time physical exercise during the past year (<1 h/w, 1 h/w, 2–3 h/w, 4–5 h/w, >5 h/w), walking/cycling (almost never, <20 min/d, 20–40 min/d, 40–60 min/d, 1–1.5 h/d, >1.5 h/d), height (splines with 2 knots), energy intake (splines with 2 knots), smoking habits (current, former, never), Charlson's weighted comorbidity index (continuous; 1–16), and diabetes mellitus (yes/no). BMI, body mass index; CI, confidence interval; HR, hazard ratio; mMED, modified Mediterranean-like diet.

Associations of cross-classified categories of BMI and mMED with total mortality are illustrated in Fig 3A, using normal BMI (mean 23 kg/m²) and high mMED as the reference. We found the lowest mortality among overweight (mean 27 kg/m²) individuals with high mMED (HR 0.94; 95% CI 0.90, 0.98). Whatever the BMI category, a high mMED score brought the point estimate of the HR to the reference level or below. In particular, obese individuals (mean BMI 33 kg/m²) with high mMED scores did not have significantly elevated HR of all-cause mortality (HR of 1.03; 95% CI 0.96–1.11). In contrast, lower BMI did not compensate for a low mMED score. No matter what the BMI, participants with a low mMED score retained an elevated risk. Indeed, participants with a normal BMI but a low mMED score had an overall mortality HR of 1.60 (95% CI 1.48–1.74), which was actually higher than that for obese individuals with high mMED (p < 0.0001 for homogeneity). We found similar estimates among women and men (Table 2) as in the pooled analysis (Fig 3A). The attenuation of the estimates after multivariable adjustment was mainly driven by differences in physical activity.

RDs and RRs are presented in S1 Table. Generally, the results followed the same pattern as that for the HRs. At 20 years of follow-up, the mortality risk difference for participants with a normal BMI and a low mMED score compared with those with a high mMED score and a normal BMI was 0.094 (95% CI 0.090–0.097), corresponding to a number needed to treat of 11 individuals.

Our secondary outcome was CVD mortality, with 12,064 cardiovascular deaths during follow-up (Fig 3B). For this outcome, the lowest mortality HRs were in participants with high mMED scores and normal or overweight BMI (Fig 3B). A high mMED score was associated with lower CVD mortality within each BMI stratum, but in contrast to findings for total mortality, individuals with high mMED scores and obesity retained a modestly elevated HR of 1.29 (95% CI 1.16–1.44). Otherwise, the patterns of the HRs were similar to those for all-cause mortality. Participants with a normal BMI but low mMED score had a CVD mortality HR of 1.76 (95% CI 1.55–1.99), which was statistically indistinguishable from the HR for the obese.

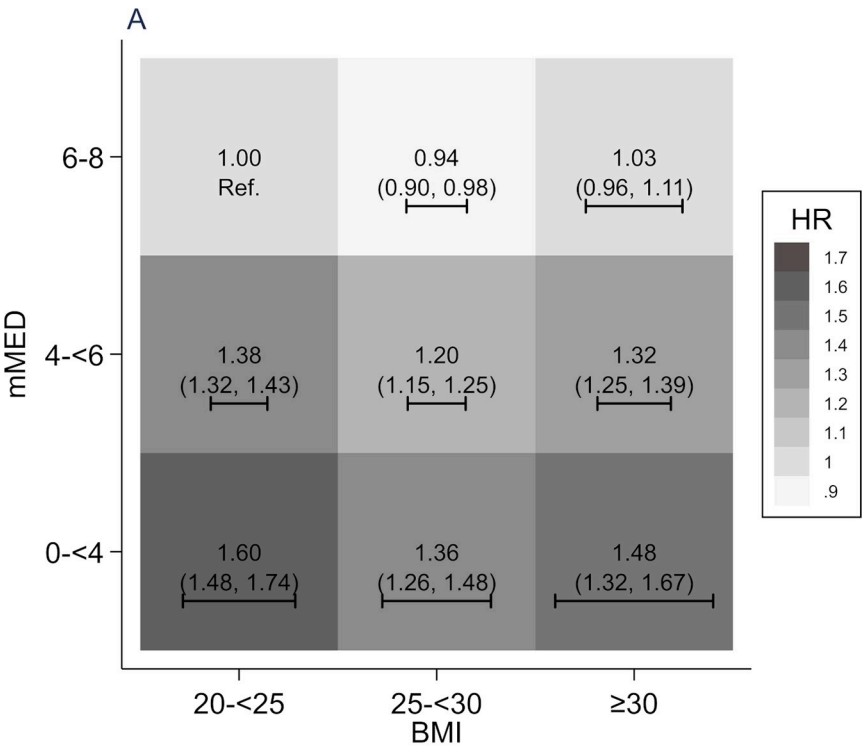

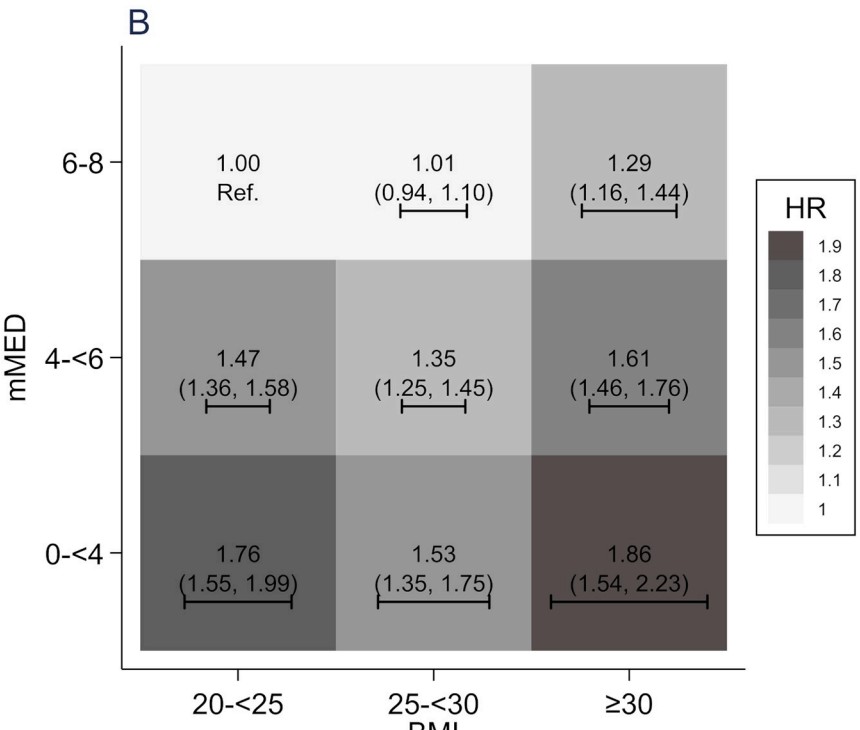

**Fig 3. Associations of combinations of BMI and adherence to an mMED with all-cause (A) and CVD mortality (B).** Estimated by multivariable-adjusted HRs by use of Cox regression analysis with a normal BMI and high adherence to mMED as the reference. The CI in each subpanel is expressed both in numbers and as a line representing the width. HRs adjusted for sex, age (splines with 2 knots), educational level (≤9, 10–12, >12 years, other), living alone (yes or no), leisure time physical exercise during the past year (<1 h/w, 1 h/w, 2–3 h/w, 4–5 h/w, >5 h/w), walking/

cycling (almost never, <20 min/d, 20–40 min/d, 40–60 min/d, 1–1.5 h/d, >1.5 h/d), height (splines with 2 knots), energy intake (splines with 2 knots), smoking habits (current, former, never), Charlson's weighted comorbidity index (continuous; 1–16), and diabetes mellitus (yes/no). BMI, body mass index; CI, confidence interval; CVD, cardiovascular disease; HR, hazard ratio; mMED, modified Mediterranean-like diet.

Further sensitivity analyses revealed similar estimates as the primary analyses for the combined exposures of BMI and mMED, including exclusion of participants with a BMI higher than 35 kg/m$^2$ (S1 Fig), restricting the analysis to those with normal BMI to a more narrow 22–25 kg/m$^2$ range (S2 Fig), additionally adjusting for pack-years of smoking (S3 Fig), restricting analysis to never-smokers (S4 Fig), and excluding individuals with any of the following criteria: ever smokers, BMI below 22 kg/m$^2$, and those with pre-existing diseases before baseline (chronic obstructive lung disease, cancer, myocardial infarction or other ischemic heart disease, heart failure, peripheral arterial disease, and stroke), as well as excluding the first 2 years of follow-up (S5 Fig).

## Discussion

In this large, population-based cohort analysis of middle-aged and older men and women, obese individuals with high adherence to a Mediterranean-type diet did not experience the increased overall mortality otherwise associated with high BMI, although a higher CVD mortality remained. However, lower BMI did not appear to counter the elevated mortality associated with a low adherence to a Mediterranean-like diet: individuals with a low mMED score retained an increased mortality even with a normal BMI. These results indicate that adherence to healthy diets such as a Mediterranean-like diet may be a more appropriate focus than avoidance of obesity for the prevention of overall mortality.

**Table 2. Combined associations of a Mediterranean diet score and BMI on all-cause mortality in women and in men.** High adherence to a Mediterranean diet and a normal body index is the reference with an HR of 1.0.

| Women | | | BMI (kg/m$^2$) | | |
|---|---|---|---|---|---|
| | | | 20.0–24.9 (mean 23) | 25.0–25.9 (mean 27) | 30 or more (mean 33) |
| Mediterranean diet score, (score points) | 6–8 (median 6.8) | Age-adjusted HR | 1.0 (ref) | 0.98 (0.92–1.06) | 1.17 (1.05–1.30) |
| | | Multivariable-adjusted* HR | 1.0 (ref) | 0.92 (0.86–0.99) | 0.99 (0.88–1.10) |
| | 4 to <6 (median 5.2) | Age-adjusted HR | 1.59 (1.49–1.69) | 1.42 (1.33–1.52) | 1.67 (1.54–1.82) |
| | | Multivariable-adjusted* HR | 1.46 (1.37–1.55) | 1.24 (1.16–1.33) | 1.31 (1.21–1.43) |
| | 0 to <4 (median 3.5) | Age-adjusted HR | 1.99 (1.76–2.26) | 1.87 (1.64–2.12) | 1.94 (1.63–2.31) |
| | | Multivariable-adjusted* HR | 1.67 (1.47–1.89) | 1.53 (1.35–1.74) | 1.46 (1.22–1.74) |
| **Men** | | | **BMI (kg/m$^2$)** | | |
| | | | 20.0–24.9 (mean 23) | 25.0–25.9 (mean 27) | 30 or more (mean 33) |
| Mediterranean diet score, (score points) | 6–8 (median 6.6) | Age-adjusted HR | 1.0 (ref) | 1.01 (0.95–1.06) | 1.30 (1.19–1.42) |
| | | Multivariable-adjusted* HR | 1.0 (ref) | 0.94 (0.89–0.99) | 1.06 (0.97–1.16) |
| | 4 to <6 (median 5.3) | Age-adjusted HR | 1.41 (1.34–1.48) | 1.33 (1.26–1.40) | 1.69 (1.58–1.81) |
| | | Multivariable-adjusted* HR | 1.32 (1.26–1.39) | 1.17 (1.11–1.24) | 1.33 (1.24–1.43) |
| | 0 to <4 (median 3.5) | Age-adjusted HR | 1.85 (1.68–2.03) | 1.61 (1.46–1.77) | 2.05 (1.76–2.39) |
| | | Multivariable-adjusted* HR | 1.57 (1.42–1.74) | 1.28 (1.16–1.41) | 1.52 (1.31–1.77) |

*Adjusted by age (splines with 2 knots), educational level (≤9, 10–12, >12 years, other), living alone (yes or no), leisure time physical exercise during the past year (<1 h/w, 1 h/w, 2–3 h/w, 4–5 h/w, >5 h/w), walking/cycling (almost never, <20 min/d, 20–40 min/d, 40–60 min/d, 1–1.5 h/d, >1.5 h/d), height (splines with 2 knots), energy intake (splines with 2 knots), smoking habits (current, former, never), Charlson's weighted comorbidity index (continuous; 1–16), and diabetes mellitus (yes/no).
**Abbreviations:** BMI, body mass index; HR, hazard ratio.

Ours is the first large cohort study examining the combined association of BMI and a Mediterranean-like diet with rates of mortality. The novelty of our study is the examination of combined strata of BMI and Mediterranean diet. The J-shaped association between BMI and all-cause as well as cardiovascular mortality confirms results from previous observational studies [2–5] as well as a mendelian randomization study based on a genetic instrument for BMI [45]. A modestly sized secondary prevention trial of a Mediterranean diet after a myocardial infarction reported more than a halved rate of all-cause mortality after 4 years among those randomized to the diet [19].

Our results are also partially consistent with those of the larger primary prevention PRE-DIMED trial. This study included 7,447 participants 55–80 years of age with a mean BMI of 30 kg/m$^2$ at high risk of CVD [20]. After 5 years of follow-up, there was a 30% reduction in risk of myocardial infarction, stroke, or cardiovascular death in those randomized to a Mediterranean diet, with an even larger effect in obese individuals. However, all-cause mortality was not affected by the intervention [20]. The magnitude of differences in the 14-point score between the Mediterranean diet intervention and the control diet group during different time points of follow-up in the PREDIMED trial was not large, ranging from 1.4 to 1.8 points, a smaller exposure contrast than in our study.

## Potential mechanisms

A high BMI has been associated with a negative impact on risk factors for premature death and CVD, including hypertension, insulin resistance, hyperlipidemia, low-grade inflammation, and oxidative stress [46]. In contrast, intervention studies have shown reduced blood pressure, improved insulin resistance, lower blood lipids, and lower inflammation and oxidative stress marker levels with Mediterranean-like diet even in those with continuing high body weight [47–53]. Additionally, these diets have effects on gut-microbiota–mediated production of metabolites influencing metabolic health [49], higher circulating adiponectin concentrations [52], and improved endothelial function [52]. Even though a Mediterranean-like diet seems to have counteracted higher all-cause mortality associated with obesity in our study, these individuals still had modestly higher CVD mortality, albeit with lower rates than obese individuals who had lower mMED scores. This remaining elevation in risk could have several different explanations; one might be the consequence of a common genetic predisposition to both high BMI and CVD [54–56]. Another biological explanation may be that an even higher adherence to a classical Mediterranean diet is needed to fully compensate for obesity or that the negative effect of obesity on cardiovascular risk factors cannot be fully compensated for by healthy eating.

The relatively high mortality rates in our study among individuals with a normal weight, even among never-smokers, might seem counterintuitive. However, nutritional reserves may be particularly needed at older ages, and sarcopenia associated with low body weight and malnutrition is a strong independent predictor of early death [57, 58]. A healthy diet, including a Mediterranean diet, is related to a lower future risk of sarcopenia, frailty, and falls [59–62]. A low BMI and a low adherence to mMED are both strongly associated with higher risk of fragility fractures [63–65], which in turn leads to high mortality rates [66, 67]. In elderly individuals, concomitant low BMI and malnutrition have also led to decreased immune function, followed by a higher risk of infections [68, 69] and higher risk of surgical complications [70], more frequent hospital admissions, and a 4-fold greater risk of mortality [71].

## Strengths and limitations

Our analysis was made possible by use of 2 population-based cohorts in a setting with wide variation in dietary habits. We had a long follow-up with a large number of deaths, ascertained

by use of national register information and personal identification numbers without loss to follow-up. We used time-updated information on diet, other lifestyle factors such as exercise and walking, socioeconomic status, and comorbidity information in our statistical analysis. Exclusion of very lean individuals from the analysis lowered the risk of reverse causation bias. The results were independent of other major known risk factors for early death, and we found consistency of the HRs in subgroups of covariates, an indication of no major confounding or effect modification. However, our results might not apply to people in other settings with different dietary patterns, to those with more extreme obesity (BMI >35 kg/m$^2$), or to younger age groups. Measurement errors in self-reported lifestyle factors such as the diet are inevitable, generally leading to conservatively biased estimates of association. Although recall of weight and height on average are quite accurate, those with high body weight tend to slightly underreport their weight [72], and therefore, some truly obese individuals might have been classified as overweight. Most importantly, our observational study of the associations of diet and BMI with mortality cannot prove that weight loss or dietary change can reduce the risk of death, and therefore, our RDs and corresponding numbers needed to treat are recommended to be cautiously interpreted. Clinical trials would be required for that level of certainty, but long-term adherence to the allocated diet is an issue with such design. Replication of our results by independent researchers and with use of other cohorts with time-updated lifestyle information would also be of additional value since recommendations cannot be based on our findings alone.

## Conclusions

The results from this longitudinal cohort study indicate that for both women and men during the last decades of life, diet can modify the association of a higher BMI with mortality; obese individuals adhering to a Mediterranean diet did not have an increased mortality in comparison to more lean individuals. In contrast, a lean BMI did not offset a poor diet. Nonetheless, a healthy diet may not completely counter higher CVD mortality related to obesity.

## Supporting information

**S1 STROBE Checklist. Checklist according to STROBE guidelines.** STROBE, Strengthening the Reporting of Observational Studies in Epidemiology.
(DOCX)

**S1 Protocol. Prospective study plan.**
(DOCX)

**S2 Protocol. Directed acyclic graph with code displaying the selection of covariates for the analysis of association of BMI combined with adherence to a Mediterranean-like diet with mortality.** BMI, body mass index
(DOCX)

**S1 Fig. Associations of combinations of BMI and adherence to an mMED with all-cause mortality after exclusion of those with BMI higher than 35 kg/m$^2$.** BMI, body mass index; mMED, modified Mediterranean-like diet
(DOCX)

**S2 Fig. Associations of combinations of BMI and adherence to an mMED with all-cause mortality after restriction of the analysis to those with normal BMI to a more narrow 22–25 kg/m$^2$ range.** BMI, body mass index; mMED, modified Mediterranean-like diet
(DOCX)

**S3 Fig. Associations of combinations of BMI and adherence to an mMED with all-cause mortality after extending the multivariable model by additional adjustment for pack-years of smoking.** BMI, body mass index; mMED, modified Mediterranean-like diet
(DOCX)

**S4 Fig. Associations of combinations of BMI and adherence to an mMED with all-cause mortality after restriction to never-smokers.** BMI, body mass index; mMED, modified Mediterranean-like diet
(DOCX)

**S5 Fig. Associations of combinations of BMI and adherence to an mMED with all-cause mortality excluding individuals with any of the following criteria: Ever smokers, BMI below 22 kg/m$^2$, and those with pre-existing diseases before baseline (chronic obstructive lung disease, cancer, myocardial infarction or other ischemic heart disease, heart failure, peripheral arterial disease, and stroke) and excluding the first 2 years of follow-up.** BMI, body mass index; mMED, modified Mediterranean-like diet
(DOCX)

**S1 Table. Associations of combinations of BMI and adherence to an mMED with all-cause mortality.** The upper part of the table presents results with use of time-updated information and 20 years of follow-up from 1997 and the lower part with use of 9 years of follow-up from 2009. The estimated associations are all multivariable-adjusted*. Absolute RDs and RRs (at 20 years and 9 years of follow-up, respectively) are calculated from the predicted survival curves based on the multivariable-adjusted Cox model. The last column of the 9 years follow-up from 2009 presents absolute RDs calculated from pseudo-observations using a GEE model with identity link. BMI, body mass index; GEE, generalized estimated equation; mMED, modified Mediterranean-like diet; RD, risk difference; RR, relative risk.
(DOCX)

## Acknowledgments

We acknowledge the Swedish Research Council-supported national research infrastructure SIMPLER for provisioning of facilities and experimental support, and we thank Anna-Karin Kolseth for her assistance. The computations were performed on resources provided by the Swedish National Infrastructure for Computing's (https://www.snic.se/) support for sensitive data (SNIC-SENS) through the Uppsala Multidisciplinary Center for Advanced Computational Science (UPPMAX) under Project SIMP2019004.

## Author Contributions

**Conceptualization:** Karl Michaëlsson.

**Data curation:** Karl Michaëlsson, Alicja Wolk.

**Formal analysis:** Karl Michaëlsson, Jonas Höijer, Bodil Svennblad.

**Funding acquisition:** Karl Michaëlsson, Alicja Wolk.

**Investigation:** Karl Michaëlsson.

**Methodology:** Karl Michaëlsson, Jonas Höijer, Bodil Svennblad.

**Project administration:** Karl Michaëlsson.

**Resources:** Karl Michaëlsson, Alicja Wolk.

**Software:** Karl Michaëlsson, Bodil Svennblad.

**Supervision:** Karl Michaëlsson.

**Writing – original draft:** Karl Michaëlsson.

**Writing – review & editing:** John A. Baron, Liisa Byberg, Jonas Höijer, Susanna C. Larsson, Bodil Svennblad, Håkan Melhus, Alicja Wolk, Eva Warensjö Lemming.

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
