## [Editor Report · Decision Letter 0]

14 Mar 2020

Dear Dr Michaëlsson, 

Thank you for submitting your manuscript entitled "Combined impact of body mass index and adherence to a Mediterranean-like diet on all-cause and cardiovascular mortality: a cohort study in women and men" for consideration by PLOS Medicine.

Your manuscript has now been evaluated by the PLOS Medicine editorial staff and I am writing to let you know that we would like to send your submission out for external peer review.

Please re-submit your manuscript within two working days.

Kind regards,

Louise Gaynor-Brook, MBBS PhD

PLOS Medicine

---

## [Decision Letter · Decision Letter 1]

2 May 2020

Dear Dr. Michaëlsson,

Thank you very much for submitting your manuscript "Combined impact of body mass index and adherence to a Mediterranean-like diet on all-cause and cardiovascular mortality: a cohort study in women and men" (PMEDICINE-D-20-00848R1) for consideration at PLOS Medicine. 

[LINK]

In light of these reviews, I am afraid that we will not be able to accept the manuscript for publication in the journal in its current form, but we would like to consider a revised version that addresses the reviewers' and editors' comments. Obviously we cannot make any decision about publication until we have seen the revised manuscript and your response, and we plan to seek re-review by one or more of the reviewers. 

We expect to receive your revised manuscript by May 25 2020 11:59PM. Please email us (plosmedicine@plos.org) if you have any questions or concerns.

We look forward to receiving your revised manuscript. 

Sincerely,

Emma Veitch, PhD

PLOS Medicine

On behalf of Clare Stone, PhD, Acting Chief Editor,

PLOS Medicine

plosmedicine.org

*In the last sentence of the Abstract Methods and Findings section, please describe the main limitation(s) of the study's methodology.

*Due to the observational design of the study, we'd suggest care in using causal language - at the moment the abstract uses language such as "reduce(s) risk..", which may be better phrased as "associated with a reduction in risk". There may be other places to modify this too.

*Did your study have a prospective protocol or analysis plan? Please state this (either way) early in the Methods section.

*We'd suggest ensuring that the study is reported according to the STROBE guideline, and include the completed STROBE checklist as Supporting Information. Please add the following statement, or similar, to the Methods: "This study is reported as per the Strengthening the Reporting of Observational Studies in Epidemiology (STROBE) guideline (SChecklist)." The STROBE guideline can be found here: http://www.equator-network.org/reporting-guidelines/strobe/. When completing the checklist, please use section and paragraph numbers, rather than page numbers.

Comments from the reviewers:

Reviewer #1: Statistical review

This paper reports the analysis of a cohort study investigating association between diet (as quantified with a Mediterranean diet score) and BMI on mortality. The follow-up is long and the sample size is high; I also thought generally the methods and reporting was good. I have some minor comments:

1. Abstract - "Obese individuals with high mMED did not experience

higher rates of all-cause mortality" - add something along the lines of 'there was no significant evidence of..' or 'did not experience significantly higher…'

2. Given the questionnaire completion was moderate, especially in men, can anything be added on how representative those who completed the questionnaire are of the general population?

3. Page 9 "Covariates, chosen using directed acyclic graphs," - can more information be given on how they were selected? Did this use any data?

4. Page 9 -how much missing data was there was on the covariates (it would be sufficient to say what proportion were complete cases)?

5. Page 9 - "We performed additional analyses in subgroups of potential confounders in which BMI below or above the median of 26 kg/m2 and mMDS were included as continuous variables.": I didn't follow this; what were the confounders, were they pre-specified and is this a stratified analysis or something different? It would be useful if an analysis plan was included if this was pre-specified.

6. Page 11 - the results of the sensitivity analyses should be provided in supplementary material (as per PLOS medicine guidance prohibiting 'data not shown').

James Wason

Reviewer #2: This well-written manuscript presents the results of individual analyses of BMI and a Mediterranean diet score, as well as a joint analysis of these, in relation to all-cause and CVD mortality in a Swedish cohort of older adults. The work has many strengths; here I will detail some areas in which I believe that it could be improved. In particular, I have grappled with what the purpose of the study is, what the results may be telling us, and what they can be used for.

Major comments:

1. The authors present a 3x3 table, detailing a joint analysis of two exposures on the multiplicative scale. This is in essence an interaction analysis, and I commend it. It appears that there is no interaction between BMI and diet quality, and that diet quality alone associates with greater all-cause mortality. The authors conclude that it is more important to focus on improving diet quality than preventing obesity to lower all-cause mortality in this age group. However, the individual analyses of BMI and diet show that each of these exposures are associated with all-cause mortality. In the absence of a multiplicative interaction, there must be an interaction on the additive scale. What are the implications of this for public health messages? Ought public health interventions to be targeting diet alone, which the conclusions of the abstract indicate should be the case? Where can most lives be saved - among those with poor diets or those who are of an unhealthy BMI (see also comment below)? If this is the purpose of the study, I would recommend redoing the analyses using methods that estimate risk differences. The pseudoobservation method can be used for time-to-event data; others also exist. (see https://doi.org/10.1186/1471-2288-14-97 ; https://doi.org/10.1515/em-2017-0015) If the intention is to assess multiplicative interactions, please be mindful of how these are greatly dependent on the baseline hazard of death, and consider this in the interpretation of the results. Also, the effects of time varying exposures on time-varying confounders, and any feedback loops between these, cannot be estimated without bias using Cox proportional hazards. In this case, g computation is preferable, as this bias can be taken into account.

2. The current conclusion of the paper, and in particular the abstract, is based on the results for all-cause mortality. The results for CVD mortality are not entirely in agreement, and given that CVD-deaths are over 1/3 of the total number of deaths, the conclusions should take these results into account. Some further discussion of the possible other important causes of death for this age group would also be useful. Are the results, which indicate that BMI is not associated with these other causes of death for a given diet quality, reasonable?

3. Further discussion of the importance of the different body compartments that BMI collectively measures in this age group would be very helpful in interpreting the results, particularly in this context of a very long follow up. The brief discussion of sarcopenia is interesting, but how does this relate to the normal BMI category? Would these participants be considered sarcopenic and frail? What are the chances that some of the obese participants actually have a high muscle mass for their age, and that this is protective? What might be considered a healthy or an unhealthy BMI in this age group, and why? While the authors explicitly state that the study cannot tell us about changes in BMI or diet quality over time, the data are available to cast some light on this…

Minor comments:

1. Please be mindful of the terms used to describe the results. For example in a couple of places, the manuscript mentions presenting rates, but the results are hazard ratios (ie a relative measure of association).

2. Please consider including the DAG used to select confounders in the supplements. For instance, I am curious how diabetes can be a (time-varying?) confounder, when this disease is partially caused by elevated BMI and poor dietary habits. I would think that diabetes was an intermediate variable, and that adjusting for this would cause bias of the associations. I am also curious as to why height is considered a confounder.

3. The purpose of the analyses presented in figure 2 is not clearly described, and the results are not interpreted in depth. Please either justify these analyses more clearly, or omit them.

4. The test used to derive the p for homogeneity is not described in the methods section.

Reviewer #3: Michaëlsson et al evaluated the combined associations of BMI and adherence to a Mediterranean-like diet on all-cause and CVD mortality among men and women in the Swedish Mammography Cohort and the Cohort of Swedish Men. When analyses were cross-classified by categories of BMI and mMED score, the authors identified the lowest rates of all-cause mortality among overweight individuals with high mMED compared with those with normal weight and high mMED. At the same time, obese individuals with high mMED did not experience higher rates of all-cause mortality while individuals with a low mMED had a high mortality despite a normal BMI. This is a well-done, interesting, and important analysis highlighting the modifying effect of diet. However, there are major methodological considerations outlined below that need to be adequately addressed:

1. Throughout the manuscript, please avoid use of causal terminology (eg: reduced � replace with "lower") given the observational nature of the study. Along these lines, it is recommended that the authors replace the term "impact" in the title to a non-causal term such as "associations".

2. This study does not aim to compare the relative effectiveness of lowering BMI versus a higher diet quality. Therefore, the "conclusion" in the abstract and the "meaning" in the key points section should reflect what the study actually did - that diet quality "modifies" the higher risk seen with higher BMI.

3. Replace the term "subjects" with "participants".

4. Respiratory diseases (such as COPD and pulmonary disease) are the major causes of low BMI. Did the authors consider excluding these individuals?

5. Several studies have shown that confounding due to smoking is strikingly strong in analyses of BMI and mortality and that complete elimination of confounding is only possible by restricting to never smokers. The current analysis remains severely confounded by smoking status which may partly explain the higher HR's in the normal BMI range compared to the obese BMI range. The authors should attempt to carefully and completely control for confounding due to smoking status.

6. Given that pack-years of smoking was available, it is not clear why this was not adjusted for in the primary analysis. Residual confounding due to smoking is inevitable, it is critical to adjust for this as completely as possible.

7. For CVD mortality analysis, did the authors exclude those with CVD at baseline?

8. The "normal" range of BMI consists of a very heterogenous group of participants including chronic smokers (see higher % of current smokers in the normal BMI category in Table 1), those with preexisting conditions, and healthy individuals. Although excluding individuals with a BMI <20 kg/m2 excludes to some extent the reverse causation due to preexisting diseases, any analyses examining BMI and mortality should consider all the following exclusions/restrictions simultaneously - 1) restrict analyses to never smokers, 2) exclude those with preexisting disease (including respiratory disease and CVD and not just cancer), 3) and exclude deaths during the first few years of follow-up. While the authors attempted to do some of these in sensitivity analyses (data not shown), it is not clear if these were done concurrently. 

9. It is recommended that the authors present analyses restricted to never smokers, normal BMI restricted to 22-25 kg/m2, excluding the first two years of follow-up and those with pre-existing diseases (respiratory diseases, cancer, CVD) as the primary analyses.

[LINK]

---

## [Decision Letter · Decision Letter 2]

30 Jul 2020

Dear Dr. Michaëlsson,

Thank you very much for re-submitting your manuscript "Combined associations of body mass index and adherence to a Mediterranean-like diet with all-cause and cardiovascular mortality: a cohort study in women and men" (PMEDICINE-D-20-00848R2) for consideration at PLOS Medicine.

I have discussed the paper with editorial colleagues and it was also seen again by two reviewers. As a result I am pleased to tell you that, provided the remaining editorial and production issues are fully dealt with, we expect to be able to accept the paper for publication in the journal.

[LINK]

Please let me know if you have any questions. Otherwise, we look forward to receiving the revised manuscript shortly. 

Sincerely,

Richard Turner PhD, for Thomas McBride, PhD

rturner@plos.org

Requests from Editors:

1- From the Financial Disclosure: “We acknowledge the national research infrastructure SIMPLER for provisioning of facilities and experimental support and we would like to thank Anna-Karin Kolseth for her

assistance.” should be placed in the Acknowledgements section at the end of the main text.

2- Please update your data statement to read: “Data cannot be shared publicly because of the sensitive nature of the data and the GDPR legislation. Data *are* available from the national research infrastructure SIMPLER for researchers who meet the criteria for access to confidential data. Details of how to obtain data from the national research infrastructure SIMPLER can be obtained at the website www.simpler4health.se” And add additional information that researchers will need to identify this specific dataset (e.g., doi or accession number).

3- Please update the title to “Combined associations of body mass index and adherence to a Mediterranean-like diet with all-cause and cardiovascular mortality: a cohort study”

4- Please include the adjustment factors in the Abstract Methods and Findings section.

5- In the Abstract Methods and Findings, please rephrase the reporting of the main outcomes to make it clear that reference for all comparisons is the normal weight and high mMED group.

6- Thank you for adding the study limitations to the Abstract. Please provide limitations that are a bit more specific to this particular study (e.g., self report, some potential sources of unmeasured confounding).

7- "... suggest that, for older women and men, ..." at the end of the abstract (note misplaced comma).

8- Thank you for including an Author Summary. Please reformat to bullet points, 1-2 sentences per bullet point, up to 3 bullets per section.

9- Thank you for including your study protocol in the supplemental files. Please also include an english translation. Additionally, the analysis plan posted at protocols.io is not publicly available, please make it so.

10- Thank you for including your STROBE statement. Please replace the page numbers with paragraph numbers per section (e.g. "Methods, paragraph 1"), since the page numbers of the final published paper may be different from the page numbers in the current manuscript.

12- I assume the questionnaires included informed consent, but please state so explicitly when describing the cohorts in the Methods section.

13- Please edit the Figure 1 legend to make clear which graphs are all-cause or CVD mortality. It would also be helpful to include labels in the figure itself. Please also describe what the shaded regions and the histogram along the bottom represent.

14- Similarly, in the Figure 2 legend, please describe what the whiskers represent (95%CIs, I presume).

15- Please include some discussion of the implications and next steps for research, clinical practice, and/or public policy just before the concluding paragraph.

16- The Article Information section should be removed from the end of the main text, and the information should appear in the relevant metadata sections via the submission form.

17- Reference 12 and some others contain competing interest information that can be cut.

Comments from Reviewers:

*** Reviewer #1: 

Thank you to the authors for addressing my previous comments well. I have no further issues to raise.

*** Reviewer #2: 

The authors have thoughtfully and thoroughly addressed my comments and concerns. The use of several methods to estimate risk differences, which all corroborate their initial findings, strengthen the manuscript (although I would have enjoyed more discussion of these findings in the Discussion section).

***

[LINK]

---

## [Editor Report · Decision Letter 3]

14 Aug 2020

Dear Professor Michaëlsson, 

On behalf of my colleagues and the academic editor, Dr. Christina Catherine Dahm, I am delighted to inform you that your manuscript entitled "Combined associations of body mass index and adherence to a Mediterranean-like diet with all-cause and cardiovascular mortality: a cohort study" (PMEDICINE-D-20-00848R3) has been accepted for publication in PLOS Medicine. 

PRODUCTION PROCESS

PRESS

PROFILE INFORMATION

Thank you again for submitting the manuscript to PLOS Medicine. We look forward to publishing it. 

Best wishes, 

Thomas McBride, PhD

Senior Editor 

PLOS Medicine

plosmedicine.org